# EPY001, a Novel Monoclonal Antibody Against *Pseudomonas aeruginosa* Targeting OprF

**DOI:** 10.3390/ijms262110380

**Published:** 2025-10-25

**Authors:** Guillaume Lacroix, Jean-Luc Lenormand

**Affiliations:** 1Epynext Therapeutics, 38000 Grenoble, France; 2Institut de Biologie Structurale (IBS), Université Grenoble Alpes (UGA), Commissariat à l’Energie Atomique et aux Energies Alternatives (CEA), Centre National de la Recherche Scientifique (CNRS), 38000 Grenoble, France; 3CNRS, UMR5525, VetAgro Sup, Grenoble INP, TIMC, University Grenoble Alpes, 38000 Grenoble, France; jean-luc.lenormand@univ-grenoble-alpes.fr

**Keywords:** monoclonal antibody, OprF, *Pseudomonas aeruginosa*, proteoliposome, Antimicrobial Resistance

## Abstract

*Pseudomonas aeruginosa* (*P. aeruginosa*) is a high-priority opportunistic pathogen responsible for severe healthcare-associated infections exhibiting multidrug resistance, emphasizing the urgent need for alternative therapeutic strategies. Monoclonal antibodies (mAbs) targeting the highly conserved outer membrane protein OprF represent a promising approach to mitigate its infectivity. OprF, the major and highly conserved outer membrane protein of *P. aeruginosa*, plays key roles in the pathogenesis of this bacterium, including biofilm formation, host cell adhesion, immune sensing, and resistance to macrophage clearance, making it a crucial factor in virulence and a promising immunotherapeutic target. Here, we report the preclinical evaluation of EPY001, an anti-OprF mAb generated by immunization of a macaque with OprF-containing proteoliposomes. EPY001 exhibited strong nanomolar binding to OprF. Epitope mapping suggests recognition of a conformational epitope, underscoring the value of proteoliposome-based immunization for membrane protein targets. Functional assays provide insights into OprF’s role in biofilm formation, pyocyanin production, and antibiotic resistance. However, in vivo studies revealed that targeting OprF alone is insufficient to protect mice from lethal infection. These findings contribute to ongoing efforts to develop effective alternatives to conventional antibiotics against this resilient pathogen.

## 1. Introduction

The World Health Organization classifies *P. aeruginosa* among the high-priority pathogens due to its resistance to carbapenems, making it a major challenge in healthcare settings, alongside *Salmonella typhi* fluoroquinolone-resistant, *Enterococcus faecium* vancomycin-resistant, and *Staphylococcus aureus* methicillin-resistant pathogens. *P. aeruginosa* is a Gram-negative opportunistic bacterium, primarily affecting individuals with compromised immune systems. *P. aeruginosa* is a major cause of nosocomial infections, including pneumonia, surgical site infections, urinary tract infections, and bacteremia. It accounts for 7.1–7.3% of healthcare-associated infections, with pneumonia being the most common. *P. aeruginosa* is responsible for 23% of Intensive Care Unit-acquired infections, primarily affecting the respiratory tract. Ventilator-associated pneumonia and healthcare-associated pneumonia due to *P. aeruginosa* have high mortality rates (32–42.8%) [1]. The limited number of new molecules with novel mechanisms of action targeting Gram-negative bacilli currently in clinical trials increases the risk of the emergence of antibiotic-resistant *P. aeruginosa* strains [2]. Given the rising threat of antibiotic-resistant *P. aeruginosa* and the limitations of existing treatments, there is a pressing need for alternative targeted strategies that can specifically neutralize the pathogen while minimizing side effects. It is therefore essential to explore alternative strategies, such as vaccines, bacteriophages, and mAbs, to combat this growing threat.

mAbs are particularly promising because they are highly specific to their target, sparing the commensal flora. They also have fewer side effects compared to antibiotics, especially fluoroquinolones and colistin, and their longer half-life offers an advantage over conventional antibiotics. Currently, there are three antibacterial mAbs, all targeting toxins. Raxibacumab [3,4,5], and obiltoxaximab [6,7,8,9] specifically target the protective antigen (PA) component of *Bacillus anthracis* toxin while bezlotoxumab binds to *Clostridium difficile* toxin B [10,11].

Several mAbs targeting *P. aeruginosa* have been developed and tested in clinical trials, but none are currently available on the market. Notable examples include MEDI3902, a bispecific mAb targeting both the Type III Secretion System (PcrV) and the exopolysaccharide (Psl) of *P. aeruginosa* (Clinical Trial Number: NCT02255760 and NCT02696902) [12,13], KB001, which targets the PcrV protein of the Type 3 Secretion System (Clinical Trial Number: NCT00638365 and NCT00691587) [14], KBPA-101, a human mAb of the immunoglobulin M isotype directed against the O-polysaccharide moiety of *P. aeruginosa* (Clinical Trial Number: NCT00851435) [15]. Additionally, there is an antibody targeting DNABII protein, aimed at collapsing *P. aeruginosa* biofilms (NCT06159725) [16] and AR-105 (Aerucin), a broadly active, fully human Immunoglubulin G (IgG) type 1 mAb directed against *P. aeruginosa* alginate (NCT03027609) [17]. Other bacterial virulence factors also present potential high-value targets for mAbs, such as the development of mAbs targeting flagellin [18] and the iron-acquisition receptor FpvA [19]. Recently, two distinct labs employed single B cell sorting from cystic fibrosis patients to isolate human mAbs targeting PcrV, a protein from the *P. aeruginosa* Type 3 Secretion System. In the first study, a one-log reduction in Colony Forming Unit (CFU)/g lung was observed in a murine pneumonia model [20], whereas the second study did not include in vivo efficacy experiments [21].

The outer membrane proteins (OMPs) of Gram-negative bacteria are considered ideal vaccine candidates because many of them expose surface epitopes that can be recognized by the host immune system. Among these OMPs, the outer membrane protein F (OprF) is the most abundant non-lipoprotein OMP in *P. aeruginosa* and has been shown to have highly conserved amino acid sequences across all 134 pathogenic and environmental strains of the bacterium [22]. It has also been shown that OprF anchors the outer membrane to the peptidoglycan layer [23,24]. Additionally, other functions have been attributed to OprF, including biofilm formation, outer membrane vesicle biogenesis, binding and adhesion to host cells, involvement in the quorum-sensing (QS) response, and sensing host immune activation through IFN-γ binding, resistance to macrophage clearance during acute infection, all of which contribute to *P. aeruginosa* virulence [25,26,27,28,29,30,31]. These findings support the potential of targeting OprF for immunotherapy development against *P. aeruginosa* infections. For example, one group is developing a broadly protective mAb targeting the OmpA family of high-copy-number outer membrane porins in Gram-negative bacteria. Using OprF as an antigen, they isolated murine memory B cells from the spleens of OprF-vaccinated mice and performed next-generation sequencing to identify unique antibodies capable of recognizing the OprF protein of *P. aeruginosa*. This antibody was shown to target *P. aeruginosa*, *Escherichia coli* (*E. coli*), and *K. pneumonia* (Mason Nunley et al., unpublished results) [32].

Developing an immunotherapy against membrane proteins such as OprF is challenging, especially due to difficulties in producing the antigen in its native conformation. Reconstituting membrane proteins in proteoliposomes for immunization is an effective strategy but overexpressing membrane proteins in cellular systems can lead to low yields and toxicity [33,34]. Cell-free protein synthesis (CFPS) offers significant advantages, bypassing toxicity and folding issues of cell-based systems. CFPS systems, such as those based on *E. coli* or wheat germ extracts, enable large-scale production and spontaneous insertion of membrane proteins into liposomes, facilitating easier reconstitution and better presentation of native antigens [34].

We hypothesize that mAbs generated by OprF proteoliposomes play a role in the bacterium’s clearance and could contribute to the prevention and treatment of infections caused by this pathogen. This hypothesis is supported by previous research demonstrating that mice immunized with OprF proteoliposome were protected against *P. aeruginosa* infection [34].

Previously, a screening of the scFv library from a macaque immunized with OprF proteoliposomes was conducted. This led to the identification of 50 anti-OprF scFv sequences, 12 of which are protected by a patent (WO 2021/013904, PCT extension PCT/EP2020/070725) [35]. In this publication, we present the results of epitope mapping, affinity, specificity, in vitro and in vivo efficacy of our mAb EPY001 against *P. aeruginosa* infection, whose scFv sequence originates from the macaque immune library.

## 2. Results

### 2.1. Affinity, Specificity and Epitope Mapping of EPY001

#### 2.1.1. Evaluation of the Binding Affinity and Specificity of the Anti-OprF mAb EPY001

The binding affinity of the anti-OprF mAb EPY001 was assessed using an ELISA-based approach on OprF-reconstituted proteoliposomes (Figure 1a).

Two WT *P. aeruginosa* strains, CHA and H103, along with their corresponding OprF-deficient mutants, CHAΔOprF and H636, were also tested in parallel (Figure 1b,c). As expected, EPY001 demonstrated strong binding to OprF proteoliposomes, with dissociation constant (Kd) values of 0.65 nM. When tested on whole bacterial cells, Kd values are 1.63 nM for CHA and 0.09 nM for H103. In contrast, no significant binding was observed with the OprF knockout strains, indicating that EPY001 specifically recognizes the OprF protein on the bacterial surface. The higher apparent affinity observed with OprF proteoliposomes likely reflects the greater abundance and accessibility of the target protein in this reconstituted system compared to intact bacterial cells. In *P. aeruginosa*, the presence of lipopolysaccharides in the outer membrane may sterically hinder antibody access to OprF epitopes, thereby reducing apparent affinity. The OprF sequence is 100% identical between the H103 and CHA strains. The difference between CHA and H103 can be explained by the fact that they are distinct strains. To evaluate the specificity of EPY001, Western blot analysis was performed on whole-cell lysates (Figure 1d). The antibody detected the OprF protein in all tested clinical *P. aeruginosa* strains (51.3B, 108.1, PA7ET, PA10ET, PA12ET and PA16ET) but showed no reactivity with lysates from clinical *Acitobacter baumannii* (*A*. *baumannii*) and *E. coli* isolates, confirming species-specific recognition of OprF by EPY001. All experiments were performed in triplicate, and results were reproducible across independent replicates.

#### 2.1.2. Epitope Mapping of the EPY001 Antibody

To determine the specific epitope recognized by the mAb EPY001, a series of extracellular loops OprF mutants were generated, each harboring modifications in one of loops 2, 4, 5, 6, 7, or 8 (see Section 4). OprF mutants’ proteins were reconstituted into proteoliposomes to recapitulate the native outer membrane topology. Binding of EPY001 to each mutant variant was then evaluated using an ELISA-based assay under native-like conditions (see Section 4.5) (Figure 2).

The results revealed a nearly complete loss of antibody binding to the OprF mutated extracellular loop 5 (mel5) variant, while partial loss of binding was observed for the OprF mel2 and mel4 variants. These findings suggest that EPY001 recognizes a conformational epitope primarily involving loop 5, with possible contributions from loops 2 and 4. This hypothesis was further supported by Western Blot analysis under denaturing conditions (see Figure A1), which demonstrated a complete loss of EPY001 binding to the OprF mel5 variant. This confirms loop 5 as the major determinant of the EPY001 epitope. Altogether, these data suggest that EPY001 targets a conformational epitope on OprF, with mel5 being the principal structural element required for binding.

### 2.2. In Vivo Evaluation of EPY001 Does Not Demonstrate Protective Efficacy Against Acute Pulmonary P. aeruginosa Infection

Given that immunization of mice with OprF proteoliposomes conferred over 80% protection against acute lung infection with the *P. aeruginosa* CHA strain [34], we sought to directly evaluate the therapeutic potential of our mAb EPY001 in vivo. In an initial curative experiment, mice (n = 6 per group) were intratracheally infected with 50 µL of a CHA bacterial suspension (5 × 10^6^ CFU/mL), followed by intraperitoneal administration of EPY001 at a dose of 20 mg/kg post-infection (Figure 3a). Tobramycin, an antibiotic to which the CHA strain is susceptible, was used as a positive control (Figure 3a).

As expected, all mice treated with tobramycin survived the infection (6/6). In contrast, only one mouse survived in the EPY001-treated group (1/6), and no survival was observed in the group receiving the control anti-lysozyme antibody (0/6), indicating limited or no therapeutic efficacy of EPY001 under these conditions. To assess whether pre-exposure to the antibody could enhance its protective effect, a second experiment was conducted in which EPY001 (20 mg/kg) was administered intraperitoneally one hour prior to infection (Figure 3b). However, no survival was observed in either the EPY001 or the control antibody groups (0/6), while all mice treated with tobramycin again survived (6/6). These results suggest that EPY001 may fail to reach effective concentrations at the site of infection, potentially due to limited pulmonary distribution. To test this hypothesis, a third experiment was performed in which EPY001 (20 mg/kg) was pre-incubated with the bacterial suspension for one hour prior to intratracheal inoculation. This approach aimed to ensure the direct delivery of antibody-bound bacteria to the lungs at the time of infection (Figure 3c). Nevertheless, no protective effect was observed under these conditions. Taken together, these in vivo experiments indicate that EPY001 does not confer detectable protection in this acute pneumonia model under the conditions tested. These findings prompted us to investigate, through in vitro assays, whether EPY001 displays subtle antibacterial activity that might require potentiation to achieve therapeutic efficacy in vivo.

### 2.3. In Vitro Efficacy of EPY001

#### 2.3.1. Deletion of OprF Alters Antibiotic Susceptibility, but EPY001 Does Not Modulate Resistance in *P. aeruginosa*

Our goal is to develop novel therapeutic strategies targeting multidrug-resistant *P. aeruginosa*. We investigated whether the antibody EPY001 could modulate the antibiotic resistance profile of *P. aeruginosa*. In collaboration with the University Hospital of Grenoble, we first evaluated the impact of OprF deletion on antibiotic susceptibility. Disk diffusion antibiograms were performed on WT CHA and CHAΔOprF (Figure 4a).

The absence of OprF in strain CHA resulted in increased resistance to ticarcillin and piperacillin. No other significant changes in susceptibility were observed, including for cefiderocol, colistin, and β-lactam/β-lactamase inhibitor combinations (ceftazidime-avibactam, imipenem-relebactam, meropenem-vaborbactam), as assessed by microdilution (see Figure A2). As CHAΔOprF shows higher resistance to ticarcillin and piperacillin, we next examined whether EPY001 could influence the resistance phenotype of the WT strain (CHA) (Figure 4b). Disk diffusion assays were performed on CHA with or without 100 µg/mL EPY001, testing susceptibility to piperacillin, ticarcillin, levofloxacin, and piperacillin-tazobactam. No change in inhibition zone diameters was observed, suggesting that EPY001 does not alter the activity of these antibiotics under standard assay conditions. Since antibody binding may be impaired in solid media, the experiment was repeated using broth microdilution with ticarcillin, with and without EPY001 (Figure 4c). Again, no difference in the minimum inhibitory concentration (MIC) was detected, confirming that EPY001 does not modulate antibiotic resistance in *P. aeruginosa* under the tested conditions. These results indicate that while OprF contributes to the intrinsic susceptibility of *P. aeruginosa* to certain antibiotics, targeting this protein with EPY001 does not affect bacterial sensitivity, further supporting its role as a neutral target in terms of resistance modulation.

#### 2.3.2. EPY001 Slightly Reduces Biofilm Formation in *P. aeruginosa* H103

To investigate the potential role of the EPY001 antibody in biofilm formation, we compared the ability of *P. aeruginosa* strain H103 and its OprF knockout mutant H636 to form biofilms over time (Figure 5a).

The CHA strain and its corresponding OprF mutant were excluded from this experiment because they are unable to produce biofilms [36]. CV staining assays revealed that the WT H103 strain exhibited significantly greater adherence and biofilm biomass at 72 h compared to the H636 mutant, indicating that OprF contributes to late-stage biofilm development. Based on this observation, we assessed whether EPY001 could influence biofilm formation in the H103 strain. Biofilms were grown for 72 h in the presence of EPY001 or a control anti-lysozyme antibody (Figure 5b). Quantification at 595 nm revealed that wells treated with EPY001 exhibited a 32% reduction on average in absorbance compared to the control antibody, suggesting a modest inhibitory effect on biofilm formation. This experiment was independently repeated four times, and consistent trends were observed, confirming the reproducibility of the findings. These results suggest that EPY001 may interfere with biofilm development, potentially through partial inhibition of OprF-dependent adhesion processes.

#### 2.3.3. EPY001 Moderately Reduces Pyocyanin Production in *P. aeruginosa* H103

To investigate the impact of the EPY001 antibody on the production of the virulence factor pyocyanin, we measured pyocyanin levels in the WT *P. aeruginosa* H103 strain and its OprF-deficient mutant H636 at multiple time points (Figure 6a).

Under the tested conditions, neither the CHA strain nor its corresponding OprF mutant produced detectable levels of pyocyanin. Quantitative analysis revealed that the H103 strain consistently produced higher levels of pyocyanin than the H636 mutant, with the most pronounced difference observed at 72 h, suggesting a role for OprF in the regulation or secretion of pyocyanin during late growth phases. Based on these observations, we next assessed whether EPY001 could modulate pyocyanin production in the H103 strain at 72 h (Figure 6b). Treatment with EPY001 resulted in a 27% reduction in pyocyanin levels compared to cultures treated with a control anti-lysozyme antibody, indicating a moderate inhibitory effect. This experiment was independently repeated four times, with consistent results across all replicates, confirming the reproducibility of the findings. Collectively, these results suggest that EPY001 may partially interfere with OprF-dependent expression of pyocyanin, pointing to its potential as a modulator of *P. aeruginosa* virulence.

#### 2.3.4. Evaluation of Complement-Dependent Cytotoxicity (CDC) Assay Activity of EPY001 on the CHA Strain

To evaluate the CDC activity of the mAb EPY001, the *P. aeruginosa* CHA strain was incubated in LB medium under three conditions: (1) with heat-inactivated guinea pig complement (dCGP), (2) with active guinea pig complement (CGP), and (3) with CGP supplemented with 50 µg/mL of EPY001 (Figure 7).

Bacterial viability was evaluated at baseline (T0) and after 90 min of incubation (T90). In the presence of dCGP, the number of viable CHA bacteria increased between T0 and T90, indicating normal bacterial growth, as expected in the absence of active complement. Conversely, when incubated with active complement alone, no increase in bacterial count was observed over the 90 min period, suggesting that the CHA strain might exhibit partial sensitivity to complement-mediated killing. Adding EPY001 to the active complement condition did not further decrease the bacterial count. The number of viable bacteria remained stable between T0 and T90, comparable to the active complement condition alone. This lack of additional reduction suggests that EPY001 does not exhibit CDC activity against the CHA strain under the tested conditions.

#### 2.3.5. EPY001 Does Not Promote Antibody-Dependent Cellular Phagocytosis (ADCP)

Given that EPY001 is a murine IgG2a isotype antibody, we sought to evaluate its potential to mediate ADCP using murine J774.1 macrophages. As a first step, we assessed whether the CHA strain *P. aeruginosa* exerts cytotoxic effects on J774.1 cells. Cytotoxicity was evaluated by measuring lactate dehydrogenase (LDH) release (Figure 8a).

Results showed that LDH levels in the presence of CHA were similar to those observed in spontaneous cell lysis controls, indicating no cytotoxicity of the CHA strain toward J774.1 macrophages. This first experiment validated the use of J774.1 cells for subsequent ADCP assays. To test for ADCP activity, bacteria were incubated with J774.1 macrophages in the presence of either EPY001 or a control anti-lysozyme antibody (100 µg/mL) for 90 min (Figure 8b). The number of non-phagocytosed bacteria remaining in the supernatant was quantified. No decrease in extracellular bacterial counts was observed for both antibody conditions relative to the initial inoculum. EPY001 did not reduce the number of extracellular bacteria compared to the control antibody. These results indicate that EPY001 does not promote phagocytic uptake by J774.1 macrophages.

## 3. Discussion

In this study, we developed and characterized EPY001, a novel mAb targeting OprF, one of the most conserved and abundant OMPs of *P. aeruginosa*. The main objective was to evaluate its therapeutic potential in vitro and in vivo. EPY001 showed high nanomolar affinity for native OprF in both proteoliposome and whole-cell assays. Functionally, the antibody partially inhibited biofilm formation and slightly reduced pyocyanin production in the H103 strain, suggesting a role of OprF in these processes. However, despite strong binding, EPY001 failed to protect mice in an acute pneumonia model, consistent with its inability to promote CDC or ADCP. The OprF mutant strain H636 exhibited a distinct antibiotic susceptibility profile compared to H103, whereas EPY001 treatment did not affect the resistance of H103. Overall, these results highlight the contribution of OprF to *P. aeruginosa* physiology and represent an initial step toward developing surface-targeting antibody therapies against this pathogen.

### 3.1. Proof of Concept for Generating High-Specificity, High-Affinity Antibodies Against Membrane Proteins via Macaque Immunization with Proteoliposomes

Affinity for both proteoliposomes and bacteria is very high [37,38]. This difference may result from the higher OprF density in proteoliposomes compared to bacterial surfaces, where the exact amount of OprF in the assay wells could not be quantified. On live bacteria, OprF accessibility might also be limited by lipopolysaccharides (LPS) and other surface-exposed proteins and could reflect variations in OprF surface expression levels or structural differences affecting epitope accessibility. For example, the CHA strain is mucoid, whereas H103 is non-mucoid. To compare affinities, Bezlotoxumab (anti-Toxin B of *Clostridium difficile*) was shown to bind to two sites in the purified recombinant toxin B with high (dissociation constant, Kd = 19 ± 5 pM) and low (dissociation constant, Kd = 370 ± 310 pM) affinity (Center For Drug Evaluation and Research; Application number: 761046Orig1s000). Interestingly, reducing affinity has been proposed to enhance the agonistic activity of immunomodulatory antibodies [39].

It has previously been shown that cross-reactivity with OprF may exist. Serum antibodies from *B. pertussis* whole-cell immunized mice bind to *P. aeruginosa* OprF, which is homologous to *B. pertussis* OmpA [40]. Our results, however, do not demonstrate cross-reactivity with *E. coli* or *A. baumannii*. However, cross-reactivity remains possible with other strains, bacterial species or SARS-CoV-2 [41,42]. Our study provides proof of concept for the generation of highly specific and high-affinity antibodies against membrane proteins following immunization of macaques with proteoliposomes.

### 3.2. Impact of OprF on β-Lactam Resistance in P. aeruginosa: Implications for Antibiotic Uptake and Membrane Permeability

Our results provide new insights into the role of OprF in the intrinsic antibiotic resistance of *P. aeruginosa*. The absence of OprF in the CHA strain increased resistance to ticarcillin and piperacillin, two β-lactam antibiotics, while no significant changes were observed for other antibiotics. Unlike OprF deletion, EPY001 treatment did not increase antibiotic resistance in the CHA strain, supporting its safety as a potential therapeutic approach. These findings align with studies by Woodruff and Hancock, suggesting that OprF is involved in antibiotic uptake as OprF mutant displays slightly increases in resistance to several β-lactam antibiotics [23,43]. Our findings raise important questions about the mechanism by which OprF modulates antibiotic susceptibility, particularly in the case of β-lactam antibiotics like piperacillin and ticarcillin. OprF may act as a channel for the influx of certain β-lactams or the anchoring function of OprF in the outer membrane might contribute to the impermeabilization of the membrane [23,43,44].

Another important consideration is the potential impact of the genetic modification used to create the CHAΔOprF mutant. The strain was generated by homologous recombination with an OprF fragment containing the *aac1* gene which encodes a gentamicin acetyltransferase-3-I (AAC(3)-I) enzyme. It is well established in the literature that gentamicin acetyltransferases of the AAC(3) family specifically modify aminoglycosides by adding an acetyl group, thereby preventing the antibiotic from binding to its target, the bacterial ribosome [45,46,47]. Furthermore, it has been reported that the aacC1 gene [48], used in this study, encodes the aminoglycoside 3-N-acetyltransferase I [AAC(3)-I], which exhibits a narrow substrate specificity and can acetylate only gentamicin, astromicin, and sisomicin [48,49]. Although this modification may contribute to resistance to certain antibiotics, it is unlikely to explain the increased resistance to piperacillin and ticarcillin. Piperacillin and ticarcillin are β-lactam antibiotics that target bacterial cell wall synthesis and do not interact with the same binding sites as aminoglycosides, it is highly unlikely that the presence of the gentamycin acetyltransferase influences resistance to these antibiotics. It is important to emphasize the need to develop novel antibacterial agents that do not promote resistance, which appears to be the case with EPY001.

### 3.3. Challenges in EPY001 Efficacy: Murine Lung Infection and FcγR-Mediated Phagocytosis

Our mouse lung infection model, based on intratracheal administration, was adapted from established protocols used to evaluate anti-*P. aeruginosa* immunotherapies such as MEDI3902 [50,51,52]. Although technically more demanding than intranasal inoculation, the intratracheal route provides better control of the infectious dose. However, this highly acute model may not fully reflect the progressive infection dynamics observed in ICU patients. Several factors could explain the lack of in vivo efficacy of EPY001. Limited antibody concentrations at the infection site due to pharmacokinetics, proteolytic degradation, or mucus sequestration could reduce local activity. The rapid and severe infection induced by strain CHA may also overwhelm early immune responses. Moreover, in vivo antigen accessibility can differ from in vitro conditions because of biofilm formation, bacterial aggregation, or conformational changes in OprF. Finally, the lack of complement or immune cell cooperation might further limit efficacy.

Unlike bispecific antibody MEDI3902 [53] or sera from OprF-I-immunized mice [54], EPY001 showed no ADCP activity, partly explaining its limited efficacy. We selected the murine IgG2a isotype, which binds to all mFcγRs and displays strong affinity for the activating receptor mFcγRI [55,56,57]. The J774 macrophage cell line was used because it is well established for antibody-dependent phagocytosis assays [58,59]. The lack of enhanced phagocytosis with EPY001, despite its high affinity and extensive bacterial coating, suggests inefficient Fc engagement and downstream signaling under our experimental conditions. Efficient FcγR activation not only depends on antibody affinity but also on surface-bound antibodies [60,61,62]. Phagocytosis is markedly reduced when antibodies are positioned more than 10 nm from the target surface [63] while higher avidity enhances phagocytic activity and bacterial killing even at lower antibody concentrations [64,65,66]. In our study, steric hindrance or limited Fc accessibility—due to epitope topology or *P. aeruginosa* surface complexity (LPS, alginate, outer-membrane proteins)—may have reduced FcγR engagement. The absence of a known positive control antibody in our ADCP assay and the lack of FcγR expression profiling in J774 cells are additional limitations. Future evaluation of EPY001 using a standardized ADCP protocol [67], particularly in the context of in vivo protection, could further clarify its effector potential.

### 3.4. Context-Dependent Role of OprF in P. aeruginosa Biofilm Development and Its Modulation by the EPY001 Antibody

The role of OprF in *P. aeruginosa* biofilm formation remains debated, with conflicting reports in the literature [68]. In our study, H103 strain produced significantly more biofilm biomass after 72 h than its isogenic OprF knockout mutant H636, suggesting that OprF contributes to biofilm maturation at later developmental stages. EPY001 treatment reproducibly reduced biofilm biomass by ~32%, supporting a functional role for OprF in adhesion and biofilm structural organization. These findings are consistent with Bukhari et al. [69], who reported reduced biofilm formation in a PAO1 ΔOprF mutant under similar conditions. A recent study [70] also showed that OprF’s contribution to biofilm formation is media-dependent: while the ΔOprF mutant formed significantly less biofilm than the WT in TSB, no difference was observed in LB medium. In contrast, Chevalier et al. [71] observed enhanced biofilm formation in the H636 mutant, describing an aggregative phenotype under aerobic conditions using LB and streptomycin-supplemented LB media over 24 h. The discrepancy with our data may be due to differences in the duration of incubation: our experiments focused on a 72 h time point, while theirs were limited to 24 h. It is possible that OprF plays distinct roles at different biofilm stages. Other studies have examined the impact of OprF under varying conditions. It has been reported that OprF is involved in biofilm formation under anaerobic conditions [72] and that more ΔoprF cells adhered to poly(dimethylsiloxane) surfaces using colony enumeration post-sonication [73]. Such discrepancies likely arise from differences in experimental conditions, incubation times, and analytical approaches. Overall, our data support a positive role for OprF in mature biofilm development under aerobic conditions in LB medium, highlighting the context-dependent and multifactorial nature of OprF function. The partial inhibition of biofilm formation by EPY001 further suggests that OprF may represent a promising therapeutic target, as also reported for anti-OmpA antibodies reducing *A. baumannii* biofilm formation [74]. One limitation of our study is that we tested a single EPY001 concentration (100 µg/mL); thus, a dose-dependent effect cannot be excluded. However, given the high affinity observed in ELISAs, this concentration likely saturates available OprF sites, making greater inhibition improbable.

### 3.5. Targeting OprF to Modulate Pyocyanin-Mediated Virulence in P. aeruginosa

Pyocyanin, a blue-green phenazine pigment produced by *P. aeruginosa*, is a major virulence factor contributing to tissue damage through oxidative stress–mediated cytotoxicity [75,76,77,78,79]. In this study, we examined the role of OprF in pyocyanin regulation and evaluated whether EPY001 could modulate this phenotype. Beyond its structural role, OprF is involved in virulence regulation, notably through its interaction with host-derived gamma interferon, which can trigger the expression of pyocyanin and the lectin PA-1L [26]. Consistent with previous reports [27,69], we observed that the OprF-deficient strain H636 produced significantly less pyocyanin than the WT H103 strain, particularly after 72 h of culture, suggesting a role for OprF in pyocyanin synthesis. Pyocyanin production is under QS control [80], and it has been proposed that OprF may transmit environmental or host-derived signals to the QS network [81]. This signaling function may be critical for synchronizing the expression of virulence factors such as elastase, exotoxin A, and lectins in response to bacterial population density. Our findings further support this hypothesis by demonstrating that EPY001 induces a modest but reproducible 27% reduction in pyocyanin production in the H103 strain at 72 h. This inhibitory effect suggests that EPY001 can interfere with OprF-mediated virulence regulation. Our results confirm the role of OprF in modulating pyocyanin production in *P. aeruginosa* and highlight the potential of targeting OprF to attenuate bacterial virulence.

### 3.6. Relevance of mAbs Targeting Bacterial Membrane Proteins in the Fight Against Infections

mAbs are established therapeutics in oncology, autoimmune diseases, and viral infections, but their use against bacterial infections remains limited. Currently approved antibacterial mAbs target bacterial toxins to neutralize them. Notable examples include bezlotoxumab, which neutralizes *Clostridium difficile* toxin B, and raxibacumab, directed against the lethal toxin of *Bacillus anthracis*. This indirect approach, which targets virulence factors rather than the bacteria themselves, aims to mitigate the pathological effects of infection without killing bacterial. Targeting bacterial membrane proteins for infection clearance is debated due to antigenic variability, regulated expression, and limited accessibility. Nonetheless, biological rationale and clinical evidence suggest potential. Vaccines offer key insights: most target capsular polysaccharides (Prevenar 20, Pneumovax, Vaxneuvance) or toxins (diphtheria, tetanus, pertussis), while newer vaccines like Bexsero and Infanrix include membrane proteins (NHBA, NadA, fHbp, PorA), demonstrating immunogenicity and clinical efficacy.

### 3.7. Exploring OprF as a Therapeutic Target in P. aeruginosa Infections: Challenges and Future Directions

Identification of novel therapeutic targets against *P. aeruginosa* remains a critical priority. OprF, a major OMP of *P. aeruginosa*, has been widely studied as a potential target for vaccine development. Several studies have explored its therapeutic potential. A recent study demonstrated that *P. aeruginosa* mutants lacking OprF exhibited significantly reduced virulence in mouse models, with 100% survival in infected animals compared to 0% survival in those infected with WT PAO1 [82]. These findings underscore the promising role of OprF in immune evasion and its potential as a target for immunotherapy. In vaccines, OprF alone induces limited protection [83] but combining OprF with adjuvants (e.g., Hydrogel) or other antigens (e.g., B subunit of heat-labile toxin, OprF-I, PcrV, flagellin B, PopB/PcrH), or advanced delivery systems (e.g., mRNA, proteoliposomes), can enhance efficacy, achieving survival rates of 50% to 100% [34,54,83,84,85,86]. However, PcrV as a single antigen conferred 56–75% protection, and adding OprF/OprI did not improve efficacy despite high antibody titers [87]. As for our results, this might be explained by the diversity of *P. aeruginosa* strains. Collectively, these publications highlight the need for multivalent immunotherapies capable of protecting against diverse strains.

EPY001 targets mel5 of OprF, which may not be optimal. For example, mice immunization with loop 8, OprI, and flagellin improved bacterial clearance [88]. Several studies show that mAbs directed against the same antigen can exhibit markedly different functional activities, depending on the specific epitope targeted [89,90,91], emphasizing that rational antibody design must consider epitope selection and accessibility. Future work should systematically compare the protective capacity of antibodies directed against different epitopes of the same antigen, in order to identify those most effective at eliciting Fc-mediated immune functions against *P. aeruginosa* and other bacterial pathogens.

A major challenge is *P. aeruginosa*’s genomic diversity, with ~74% of its pangenome uncharacterized (UniProt), complicating universal target identification [92]. AI and machine learning can help analyze omics data to identify complementary targets. Future strategies include antibody–drug conjugates (ADCs) combining specific bacterial targeting with bactericidal peptides [93], as well as Fc engineering or afucosylation to enhance ADCC and phagocytosis [94,95,96,97,98]. However, strong preclinical opsonophagocytic activity often fails to translate to humans [99], and opsonophagocytosis alone is not a reliable efficacy predictor [100]. These observations highlight the complexity of developing effective antibacterial antibodies and support the need for multifaceted design strategies beyond traditional mechanisms.

## 4. Materials and Methods

### 4.1. Bacterial Strains and Growth Conditions

The strains were *P. aeruginosa* H103 (PAO1 WT prototroph); its *oprf* mutant H636 obtained by homologous recombination with an OprF fragment containing a streptomycin (Sm) cassette [43]; the mucoid strain CHA (isolated in 1990 from the broncho-pulmonary tract of a cystic fibrosis patient at the University Hospital of Grenoble) and its *oprf* mutant CHA ∆OprF strain. All these strains were kindly given by Pr B. Toussaint. *P. aeruginosa* was cultured in Luria–Bertani (LB) broth and agar (Sigma-Aldrich, St. Louis, MO, USA) without antibiotics at 37 °C for 24 h under agitation at 280 rpm. The bacteria were stored at −80 °C in vials containing LB broth supplemented with 16% glycerol. Additional bacterial strains were used to assess assay specificity (Appendix A).

### 4.2. Production and Purification of OprF Proteoliposomes

Liposomes were prepared by initially drying a lipid mixture in chloroform (Lipid Composition: cholesterol, 1,2-dioleoyl-sn-glycero-3-phosphocholine [DOPC], 1,2-dioleoyl-sn-glycero-3-phosphoethanolamine [DOPE], 1,2-dimyristoyl-sn-glycero-3-phosphate [sodium salt] [DMPA], molar ratio [2-4-2-2]. (Avanti Polar Lipids, Alabaster, AL, USA) by evaporation under a nitrogen stream. The resulting lipid film is frozen at −20 °C. The lipidic film was hydrated in 500 μL of Tris buffer (50 mM, pH 7.5) by pipetting and vortexing followed by four freeze/thaw cycles in −80 °C. The lipidic mixture was extruded using an extruder (Avanti Polar Lipids) to produce liposomes with an average diameter of 200 nm. Liposomes were stored at 4 °C.

Molecular cloning of WT and mutated OprF DNA sequences has been performed by Proteogenix^®^ (Strasbourg, France) in the pIVEX2.4d, which is a DNA plasmid optimized for the Cell-free expression of recombinant proteins in an *E. coli* cell lysate (Table 1 and Appendix A). The full length OprF from *P. aeruginosa* was then synthesized in presence of liposomes using a CFPS kit (RTS™ 500 ProteoMaster™ *E. coli* HY Kit, biotechrabbit^®^, Berlin, Germany). The recombinant plasmid pIVEX2.4-OprF [34] and the liposomes were combined with the cell lysate and the reaction mixture at a final concentration of 15 μg/mL and 4 mg/mL, respectively. The resulting proteoliposomes were then purified by sucrose-gradient ultracentrifugation (ThermoFisher SORVALL WX90+ Ultra Series, Waltham, MA, USA). The cell-free reaction mixture was first loaded on top of a 10 mL 0–40% sucrose gradient in Tris buffer (50 mM, pH 7.5) and the tube was centrifuged at 41,000 rpm (4 °C) for 2 h. 1 mL fractions were recovered from the top to the bottom of the gradient and analyzed by Western blot which was developed with an anti-6-His antibody coupled with HRP (Sigma-Aldrich, St. Louis, MO, USA). Tris buffer (50 mM, pH 7.5) was then added to each fraction containing OprF proteoliposomes and the solution was centrifuged at 30,000× *g* for 30 min at 4 °C to pellet proteoliposomes. The pellet was washed twice for 30 min at 4 °C and then resuspended in a convenient volume of Tris buffer (50 mM, pH 7.5). The purity and the protein concentration of OprF proteoliposomes were determined on a Coomassie stained SDS–PAGE gel.

### 4.3. Coomassie Stained SDS–PAGE Gel

OprF concentration was obtained by comparison to a range of known BSA concentrations. The different samples, previously diluted in 4X Laemmli buffer and heated at 95 °C for 5 min, were loaded onto a 12% acrylamide gel at 10 µL per well. Electrophoresis was carried out at 150 V in 1X TG-SDS buffer for 1 h and 30 min. The gel was stained using Coomassie blue during 24 h. After washing, gel was pictured using the ChemiDoc XRS+ camera (BIO-RAD, Hercules, CA, USA).

### 4.4. Production and Purification of Antibodies

To evaluate EPY001 in a murine model, we chose to produce and purify the antibody in the IgG2a isotype format in order to maximize the likelihood of in vivo efficacy. The IgG2a isotype is known to have high affinity for murine Fc gamma receptors, including FcγRI, FcγRIIB, FcγRIII, and FcγRIV. In contrast, the IgG1 isotype shows poor binding to FcγRI and FcγRIV. This selection was made to increase the probability that EPY001, once bound to its bacterial target, would be effectively recognized by immune effector cells expressing these receptors. For instance, murine neutrophils do not express FcγRI but do express FcγRIV, making IgG2a particularly suitable for engaging this cell population [101,102,103]. As a control antibody, we used a chicken lysozyme-specific IgG2a. Production and purification of both antibodies were performed by Icosagen^®^ (Tartu, Estonia). The murine IgG2a recombinant antibody EPY001 and the IgG2a isotype control (anti-chicken lysozyme, see Appendix A) were produced via transient expression in mammalian cells. Heavy and light chain coding sequences were cloned into pLIC1.1 expression vectors (Icosagen^®^), and all constructs were sequence-verified prior to transfection. Transient expression was performed in CHOEBNALT85 cells (1 × 10^9^ cells per batch) cultured in CHO TF medium (Xell AG, Duisburg, Germany). Chemical transfection was carried out using the R007 reagent. Production volumes were 900 mL for EPY001 and 400 mL for the isotype control antibody. Transfection quality control was conducted 96 h post-transfection by PCR and sequencing of plasmid DNA. Antibody expression and secretion were assessed by SDS-PAGE with Coomassie staining. Productivity estimates were obtained from endpoint gel analyses. Clarified culture supernatants were subjected to Protein A affinity purification using HiTrap MabSelect PrismA columns (Cytiva, Marlborough, MA, USA), followed by preparative size-exclusion chromatography (SEC). Final formulations were sterile filtered through 0.22 µm filters (Merck Millipore, Darmstadt, Germany). Quality control of purified antibodies included SDS-PAGE (SimplyBlue SafeStain) and analytical SEC (Waters BioSuite 250, 4 µm UHR SEC, 4.6 × 300 mm). Both EPY001 and the control antibody displayed a monomer content > 99% and endotoxin levels < 0.01 EU/mg, as determined using the Endosafe Nexgen-MCS LAL system (Charles River, Wilmington, NC, USA). Antibodies were formulated in PBS (pH 7.4) and stored at −80 °C.

### 4.5. Affinity of the mAb EPY001

To determine the affinity of the mAb EPY001 for different *P. aeruginosa* strains (CHA, CHA ∆OprF, H103 and H636) and OprF proteoliposomes, enzyme-linked immunosorbent assays (ELISA) were performed. Bacterial cultures were grown to an optical density of 2, then washed and resuspended in PBS before 200 µL was fixed overnight at 4 °C at the bottom of a Maxisorp plate (NUNC Maxisorp, ThermoFisher^®^, Waltham, MA, USA) under agitation (300 rpm). OprF proteoliposomes were used at 100 ng/mL in a volume of 200 µL per well for ELISAs. These plates are designed to bind both hydrophilic and hydrophobic components. Next, a blocking solution composed of 1X TBS–0.1% Tween 20 (TBST) and 5% milk was applied for two hours to prevent nonspecific binding. This step, as well as all subsequent steps, was carried out at room temperature under agitation (300 rpm). The plate was then washed with 0.1% TBST to remove any unbound material. A series of 10-fold serial dilutions of EPY001 was then prepared, starting from an initial concentration of 30 µg/mL in 0.1% TBST + 5% milk. The antibody was added to the wells at a volume of 200 µL per well and incubated for one hour. After washing, 200 µL of a secondary anti-mouse antibody conjugated to HRP (ThermoFisher ECL™ Anti-mouse IgG, Horseradish Peroxidase-linked F(ab’)2 fragment (from goat) was added to each well at a 1:1000 dilution in 0.1% TBST + 5% milk. Another one-hour incubation was performed. For detection, the HRP substrate TMB (3,3′,5,5′–Tetramethylbenzidine, Sigma-Aldrich) was used. During the enzymatic reaction, a blue coloration appeared in the wells where the secondary antibody had bound. To stop the reaction between HRP and TMB, an acidic solution (1M HCl) was added after 10 min. Absorbance was measured at 450 nm and 550 nm using a TECAN^®^ plate reader (Männedorf, Switzerland). Absorbance data as a function of molar concentration were then entered into the Prism^®^ software (GraphPad v9.2.0), with the absorbance values of bacteria alone subtracted. Using the software’s “One site–Specific binding” function, the Kd constant was calculated. The Kd range corresponds to the 95% confidence interval in which the determined Kd value falls.

### 4.6. Animal Investigation Protocol

The mouse acute pneumonia and systemic model was reviewed and approved by the University Grenoble-Alpes Animal Care Committee (approvals: APAFIS #37829-2022062911214413 v5) and conducted in an animal biosafety level 2 facility at the University of Grenoble-Alpes, France. Pathogen-free male C57BL6J mice (8 weeks old, 20 g; Janvier) were used in all animal studies. Mice were housed in a climate-controlled housing room with a daily 12 h light and 12 h dark cycle. They were provided a standard chow diet and water ad libitum. Acclimatization period of mice was one week in accordance with animal welfare recommendations.

### 4.7. Mice Acute Pneumonia Model

Mice (n = 6 per experimental group) were randomized to receive an intraperitoneal injection (100 µL) of either 20 mg/kg tobramycin, 20 mg/kg anti-lysozyme IgG control, or 20 mg/kg EPY001. Treatment was administered either one hour before (prophylactic setting) or one hour after (curative setting) bacterial challenge. Prior to bacterial instillation, mice were anesthetized via intraperitoneal injection of 37.5 mg/kg ketamine (Imalgene 1000) combined with 0.05 mg/kg medetomidine (Domitor, Orion Pharma, Turku, Finland). Pneumonia was induced by delivering 50 µL of a bacterial suspension containing 1 × 10^8^ CFU/mL of *P. aeruginosa* CHA strain (=5 × 10^6^ UFC) directly into the lungs through a 22-G catheter (BD Insyte, Franklin Lakes, NJ, USA). The catheter was removed immediately following the instillation. Following endobronchial bacterial challenge, rectal temperature, clinical signs of pain, pulmonary dysfunction, and body weight were monitored three times daily. Mice presenting a respiratory rate greater than 75 breaths per minute, cyanosis, coughing, weight loss exceeding 20%, or a rectal temperature below 30 °C were euthanized according to humane endpoint criteria. Mice surviving until the end of the 3 days post-infection period were euthanized. All clinical assessments were performed by investigators blinded to the treatment groups.

Given the severity of the challenge, a third experiment was conducted in which bacteria were pre-incubated with EPY001 for one hour prior to intratracheal administration. This approach aimed to ensure both effective binding of EPY001 to the bacteria and its presence in the lungs before it became too late to achieve therapeutic efficacy.

### 4.8. Antibiotic Susceptibility Testing

Antibiogram and microdilution for antibiotic susceptibility testing were performed according to the European Committee on Antimicrobial Susceptibility Testing (EUCAST). The CHU Grenoble Alpes performed antibiograms using Bio-Rad^®^ antibiotic disks. MICs for ceftazidime-avibactam, imipenem-relebactam, and meropenem-vaborbactam were determined using ETEST^®^ (bioMérieux, Marcy-l’Étoile, France). The CHU Grenoble Alpes also evaluated the clinical performance of Bruker’s UMIC^®^ cefiderocol and colistin by determining MICs via microdilution using the WT strain CHA and CHAΔOprF. Disk diffusion antibiogram and MICs determination comparing *P. aeruginosa* CHA and CHAΔOprF was performed once by the CHU of Grenoble.

We performed antibiogram on CHA with and without EPY001. Briefly, MH agar plates were inoculated with bacterial suspensions (0.5 McFarland) with or without a pre-incubation (30 min 37 °C) with 100 µg/mL EPY001. Commercial antibiotic disks (Ticarcillin, Piperacillin, Levofloxacin and Piperacillin/Tazobactam, Oxoid, Thermo Fisher, Waltham, MA, USA) were placed on the agar surface. The amounts of antibiotic in the diffusion disks are as follows: Ticarcillin, 75 µg; Piperacillin, 30 µg; Levofloxacin, 5 µg; and Piperacillin/Tazobactam, 30 µg/6 µg. Minimal Inhibitory Concentration (MIC) Determination values were determined using the broth microdilution method in 96-well plates, following EUCAST recommendations. Serial two-fold dilutions of antibiotics were prepared in MH broth, and bacterial inocula (~5 × 10^5^ CFU/mL) were added to each well. Plates were incubated at 37 °C for 16–20 h. MIC was defined as the lowest antibiotic concentration that completely inhibited visible bacterial growth. The concentration range of ticarcillin used was 0–100 mg/L. All experiments comparing *P. aeruginosa* CHA with or without EPY001 were conducted in triplicate by the authors. LB with no bacteria was used as a negative control. It allows us to detect the potential contamination of our samples.

### 4.9. Biofilm Formation Assays

*P. aeruginosa* was grown overnight in LB broth at 37 °C. Next day the cultures were diluted in LB to 10^7^ cfu/mL and dispensed into a transparent untreated sterile 24-well flat bottom plate (SPL) with or without 100 µg/mL EPY001. The microtiter plates were incubated at 37 °C for 24 h, 48 h and 72 h. After that the cell suspension was removed and the plates were washed twice with 0.9% NaCl and inverted to dry at room temperature for 1 h. Following this 150 µL of 0.1% CV solution (Thermo Scientific Chemicals, Waltham, MA, USA) was added to the wells and was allowed to stain for 15 min. After staining, CV was removed, and the wells were washed 3 times with 0.9% NaCl. The bound CV was then solubilized by adding 200 µL of absolute ethanol. The absorbance of CV was read at 595 nm using a TECAN^®^ plate reader. Experiments were conducted in quadruplicate. Blank samples (and negative control) is composed of LB no bacteria. These samples allow the subtraction of background absorbance from the rest of the data points to ensure the most accurate OD readings.

### 4.10. Pyocyanin Production Assays

*P. aeruginosa* was grown overnight in 5 mL of LB medium with or without 100 µg/mL EPY001 or IgG control at 37 °C with orbital shaking at 180 rpm during 72 h. A total of 5 µL of an overnight culture were used as inoculate (dilution 1:1000^e^). After centrifugation at 10,000× *g* for 10 min, the phenazine pigment was extracted from the supernatant with 3 mL of chloroform and 0.2 N HCl. The amount of pyocyanin was estimated as the ratio of the absorbance at 520 nm to that at 600 nm. Experiments were conducted in quadruplicate.

### 4.11. CDC Activation Assay

Exponential-phase *P. aeruginosa* were centrifuged and the pellet resuspended with PBS. Using a previously established bacterial growth curve, the initial bacterial quantity was determined, and the necessary dilution was performed to obtain a final concentration of 1 × 10^6^ CFU/mL. The diluted bacteria were then placed in a 96-well cell culture plate (CELLSTAR Greiner Bio-One, 96-well cell culture plate, Frickenhausen, Germany) and incubated with the EPY001 antibody at a final concentration of 50 µg/mL for 30 min at room temperature to allow antibody binding. Finally, eather 40% of active CGP (Merck KGaA, Complement, Guinea Pig Serum, Darmstadt, Germany) or 40% of heat-inactivated complement (56 °C, 30 min) were added. The plate was shaken for 30 s and incubated at 37 °C without shaking for 90 min. CFUs were enumerated by plating serial dilutions (10^−2^ to 10^−6^) on LB agar without antibiotics at T0 and T90. Experiments were conducted in quadruplicate and CFU counting was performed in triplicates.

### 4.12. Cytotoxicity Assay (LDH Release)

Lactate dehydrogenase (LDH) release was measured using the CyQUANT™ LDH Cytotoxicity Assay Kit (Invitrogen, Carlsbad, CA, USA), following the manufacturer’s instructions. J774 macrophages (10,000 or 20,000 cells per well) were incubated for 2, 4, and 6 h with strains CHA, CHA ∆OprF, H103, and H636 at a multiplicity of infection (MOI) of 10, in the presence or absence of EPY001 antibody at a concentration of 100 µg/mL.

### 4.13. ADCP Activation Assay

J774A.1 murine macrophages were kindly given by Jean-Pierre Alcaraz. Cells were maintained in DMEM (Gibco) supplemented with 10% Fetal Bovine Serum (FBS, Gibco, Carlsbad, CA, USA) and penicillin-streptomycin (P/S, Gibco) at 37 °C with 5% CO_2_. One day prior to the assay, cells were split 1:2. On the day of the experiment, macrophages were detached using trypsin, washed twice in PBS, and resuspended in DMEM without FBS or antibiotics supplemented with 1% Bovine Serum Albumin (BSA, Gibco). Cell concentration was determined using trypan blue exclusion and a KOVA counting chamber and adjusted to 1 × 10^6^ cells/mL. *P. aeruginosa* strains CHA and PAO1 were grown from overnight precultures in LB medium (1:10 dilution) at 37 °C and 280 rpm to mid-log phase. Bacteria were harvested by centrifugation (5 min, 5000× *g*), washed twice with sterile PBS, and resuspended in DMEM without FBS or antibiotics containing 1% BSA. The bacterial suspension was adjusted to an OD_600_ of 0.4 (~1 × 10^8^ CFU/mL). Macrophages (317.5 µL; 1.57 × 10^6^ cells/mL) were mixed with bacteria (25 µL; 2 × 10^8^ CFU/mL) in 2 mL tubes. Either 150 µL of CGP Serum (Merck KGaA, Darmstadt, Germany) or dCGP Pig Serum was added, followed by 7.5 µL of either EPY001 (5 mg/mL) or anti-lysozyme IgG control. Samples were incubated at 37 °C. Total volume of the final reaction components was 500 µL. Aliquots were collected at baseline (T0) and after 120 min (T120) for CFU enumeration. Serial dilutions (10^−3^ to 10^−6^) were prepared in PBS and plated in duplicate on LB agar. Plates were incubated overnight at 37 °C, and CFUs were enumerated. The following groups were tested: CHA + IgG control and CHA + EPY001. Experiments were conducted in triplicate.

### 4.14. Statistical Analysis

Nonparametric tests were performed for all statistical analyses using the R software environment, version 3.6.3. Two-sided Wilcoxon–Mann–Whitney tests were used for all experiments with the exception of the analysis presented in Figure 5b and Figure 6b, in which the one-sided Wilcoxon–Mann–Whitney test was used because we had an a priori hypothesis that the anti-OprF antibody would specifically reduce biofilm formation and pyocyanin production compared to the control. A *p* value ≤0.05 was considered to be significant.

## 5. Conclusions

In this study, we report the successful generation of high-affinity mAb EPY001 targeting the OprF of *P. aeruginosa* using a proteoliposome-based immunization strategy in non-human primates. EPY001 demonstrated strong binding specificity toward clinical *P. aeruginosa* strains and was associated with a reduction in both biofilm formation and pyocyanin production—highlighting a potential role for OprF in these critical virulence mechanisms. However, EPY001 failed to show sufficient in vivo efficacy in a model of acute lung infection, and no Complement-dependent cytotoxicity or Antibody-Dependent Cellular Phagocytosis activity was detected. Our findings indicate that targeting OprF alone is unlikely to yield therapeutic efficacy against all *P. aeruginosa* strains’ infections. Future therapeutic strategies should consider multi-target antibody approaches to counteract the multifactorial resilience of *P. aeruginosa* in acute and chronic infections.

## Figures and Tables

**Figure 1 ijms-26-10380-f001:**
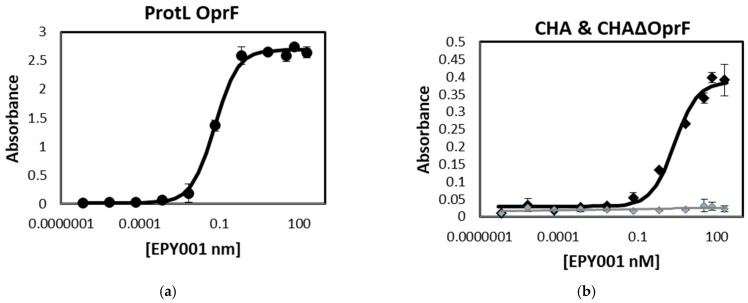
EPY001 mAb specifically binds to OprF protein from *P. aeruginosa*. (**a**) ELISA binding curves of EPY001 to OprF-reconstituted proteoliposomes; (**b**) binding affinity of EPY001 to whole cells of Wild-Type (WT) *P. aeruginosa* strains CHA and its corresponding OprF-deficient mutants CHAΔOprF; (**c**) binding affinity of EPY001 to whole cells of WT *P. aeruginosa* strains H103 and its corresponding OprF-deficient mutants H636. Kd values were determined using nonlinear regression analysis (one site—specific binding) in GraphPad Prism v9.2.0. All data are representative of at least three independent experiments; (**d**) Western blot analysis of whole-cell lysates using EPY001 as the primary antibody (IgG2a), followed by a horseradish peroxidase (HRP)-conjugated goat anti-mouse IgG secondary antibody.

**Figure 2 ijms-26-10380-f002:**
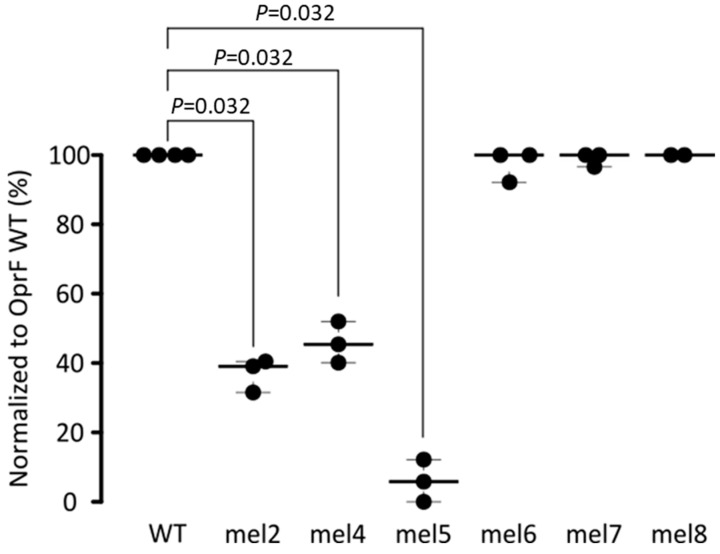
Binding of EPY001 to WT and mutant OprF proteoliposomes assessed by ELISA. EPY001 Epitope mapping was evaluated by ELISA on proteoliposomes reconstituted with either WT OprF or mutant variants carrying targeted modifications in extracellular loops 2, 4, 5, 6, 7, or 8 (n = 3). Results are expressed as percentage of binding relative to WT OprF proteoliposomes (set at 100%). Statistical analysis was performed using the Mann–Whitney U test; differences were considered significant for *p* < 0.05.

**Figure 3 ijms-26-10380-f003:**
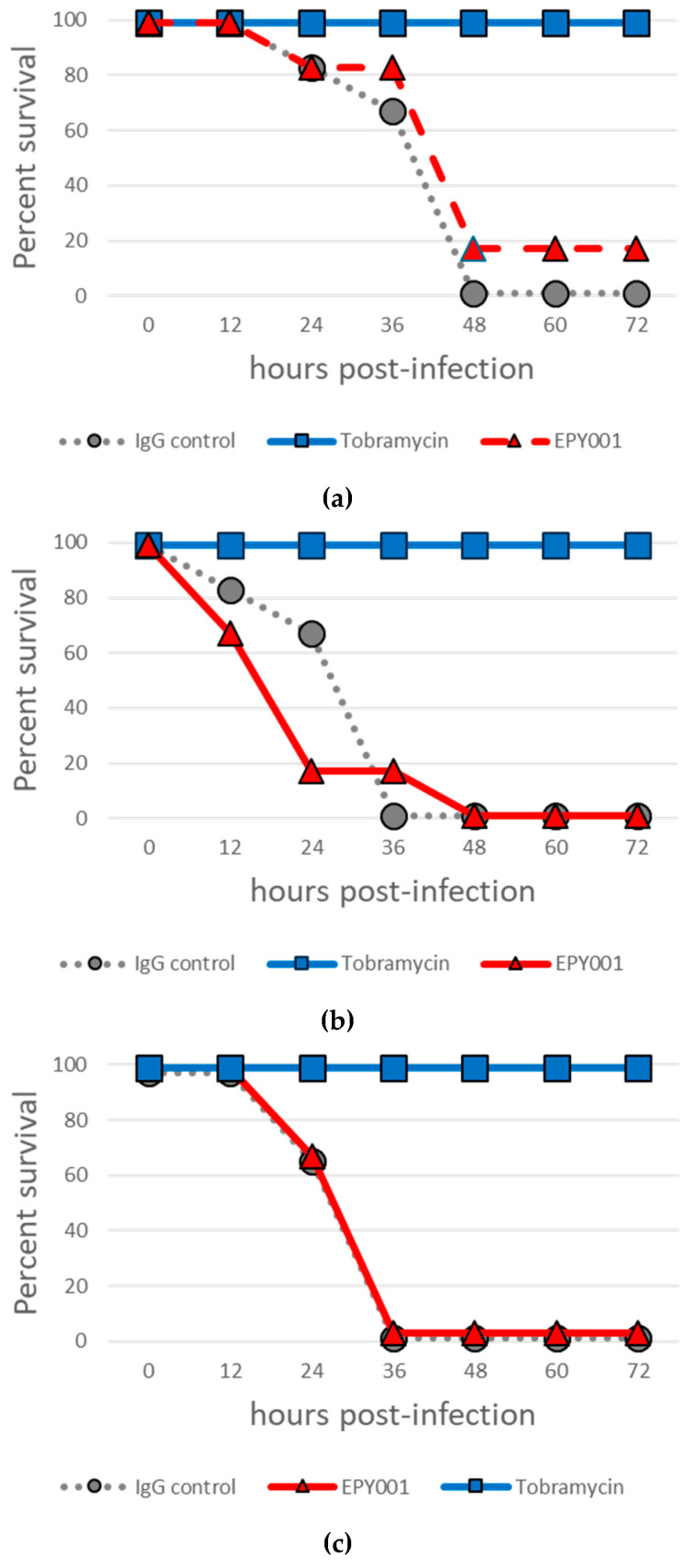
Passive immunization with EPY001 fails to confer protection in a murine model of acute *P. aeruginosa* pneumonia. C57BL/6J mice (n = 6 per group) were challenged via the orotracheal route with 50 µL of a bacterial suspension containing 1 × 10^8^ CFU/mL of *P. aeruginosa* CHA strain (final dose: 5 × 10^6^ CFU per lung). Three independent survival experiments were performed over a 3-day period: (**a**) curative setting—EPY001 (20 mg/kg), control IgG (20 mg/kg), or tobramycin (20 mg/kg) were administered intraperitoneally after infection; (**b**) prophylactic setting—same treatment groups, with antibodies/antibiotic administered intraperitoneally 1 h before infection; (**c**) pre-opsonization setting—bacteria were pre-incubated with EPY001 for 1 h prior to orotracheal infection, while control IgG and tobramycin were administered intraperitoneally. Survival was monitored for 72 h post-infection.

**Figure 4 ijms-26-10380-f004:**
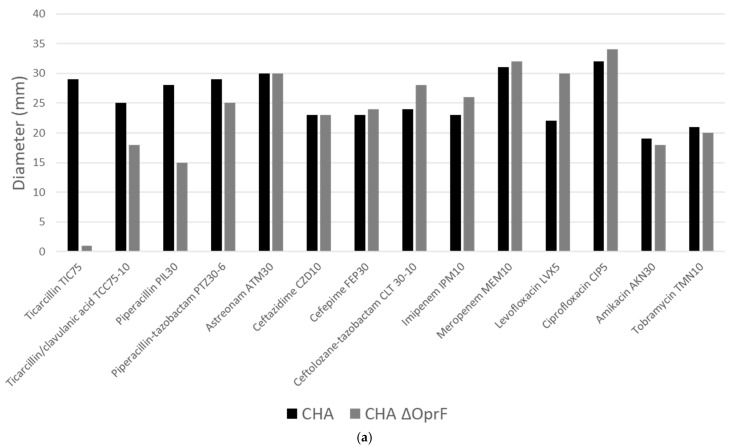
Effect of the EPY001 antibody on antibiotic susceptibility of *P. aeruginosa* CHA. (**a**) Disk diffusion antibiogram comparing *P. aeruginosa* CHA and CHAΔOprF. The amounts of each antibiotic are indicated on the texte in µg. For example Levofloxacin LVX5 indicates 5 µg; (**b**) disk diffusion antibiogram comparing Ticarcillin, Piperacillin, Levofloxacin, and Piperacillin/Tazobactam against CHA, CHA treated with 100 µg/mL of the EPY001 antibody, and CHAΔOprF; (**c**) minimum inhibitory concentration (MIC) of Ticarcillin determined by microdilution for CHA alone, CHA + EPY001 (100 µg/mL), and CHAΔOprF.

**Figure 5 ijms-26-10380-f005:**
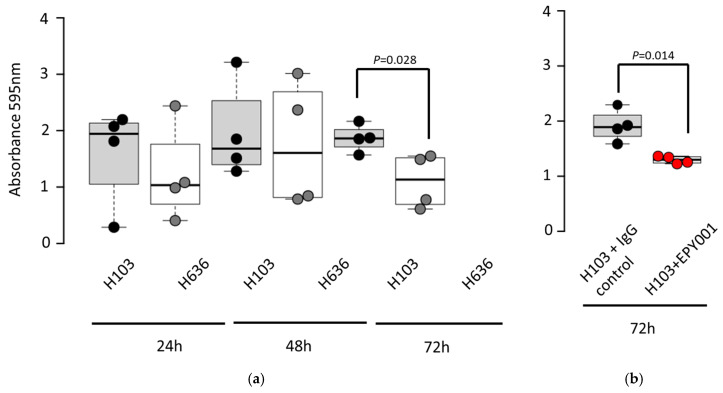
EPY001 antibody reduces *P. aeruginosa* H103 biofilm formation. (**a**) Biofilm formation by WT H103 and its ΔOprF mutant H636 was measured over 72 h using crystal violet (CV) staining; statistical significance was assessed using the bilateral Mann–Whitney test, with *p*-values < 0.05 considered significant; (**b**) biofilms of H103 were grown for 72 h in the presence of EPY001 or a control antibody (100 µg/mL); a one-tailed Mann–Whitney test was used because we had an a priori hypothesis that the anti-OprF antibody would specifically reduce biofilm production compared to the control with *p*-values < 0.05 considered significant. Experiments were independently repeated four times.

**Figure 6 ijms-26-10380-f006:**
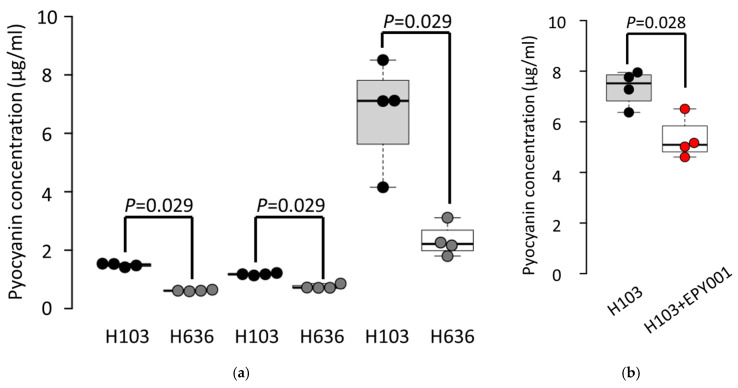
Effect of OprF and EPY001 on pyocyanin production in *P. aeruginosa*. (**a**) Quantification of pyocyanin production by the WT *P. aeruginosa* H103 strain and its OprF-deficient mutant H636 at multiple time points (24, 48, and 72 h). Pyocyanin levels were measured from culture supernatants using a standard chloroform-HCl extraction protocol; statistical significance was assessed using the bilateral Mann–Whitney test, with *p*-values < 0.05 considered significant; (**b**) pyocyanin production by H103 cultures treated for 72 h with 100 µg/mL of EPY001 or control human IgG (anti-lysozyme); EPY001 treatment resulted in a 27% decrease in pyocyanin concentration compared to the control; a one-tailed Mann-Whitney test was used because we had an a priori hypothesis that the anti-OprF antibody would specifically reduce pyocyanin production compared to the control with *p*-values < 0.05 considered significant. Data represent four independent experiments.

**Figure 7 ijms-26-10380-f007:**
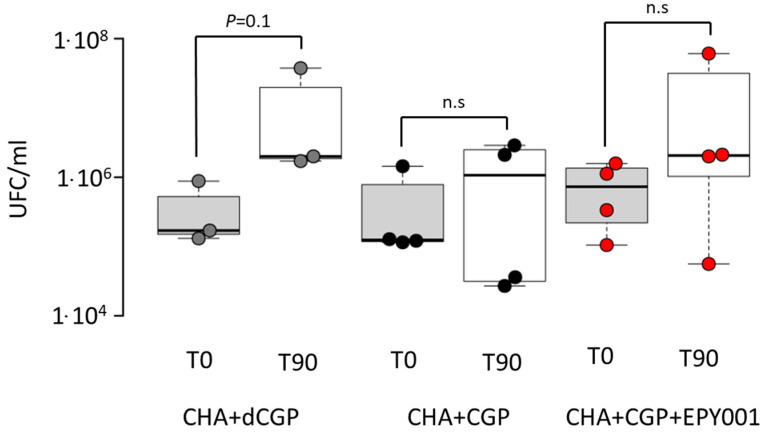
CDC activity of the EPY001 mAb against *P. aeruginosa* CHA strain. Bacterial viability was assessed after incubation of the CHA strain in LB medium under three conditions: (1) with dCGP, (2) with CGP, and (3) with CGP supplemented with EPY001 (50 µg/mL). Bacterial counts were measured at T0 and after 90 min of incubation (T90). Data represent three independent experiments. Statistical comparisons between T0 and T90 were performed using the Mann–Whitney U test; *p*-value ≤ 0.05 was considered statistically significant; n.s, for not significant.

**Figure 8 ijms-26-10380-f008:**
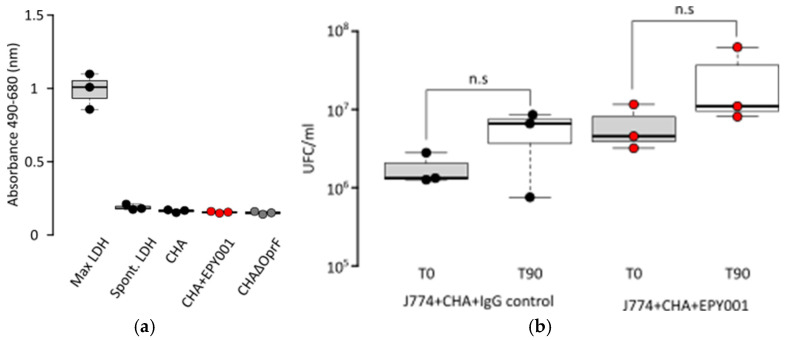
Evaluation of cytotoxicity and ADCP activity of EPY001 using J774.1 murine macrophages. (**a**) Cytotoxicity of the *P. aeruginosa* CHA strain toward J774.1 macrophages was assessed by measuring LDH release after 90 min of co-culture. Absorbance was recorded at 490–680 nm; (**b**) ADCP assay. J774.1 macrophages were incubated with the CHA strain and either EPY001 or a control anti-lysozyme IgG2a antibody (100 µg/mL) for 90 min. The number of non-phagocytosed bacteria remaining in the supernatant was quantified at T0 and T90. Data represent three independent experiments. Statistical analysis was performed using the Mann–Whitney test; differences were considered significant at *p* < 0.05; n.s, for not significant.

**Table 1 ijms-26-10380-t001:** Mutation sites in the extracellular loops of OprF.

Mutated Extracellular Loop OprF	Mutations
mel2	Y94A E95A N98A K100A
mel4	D180A H183A Q184A E1861
mel5	D202A D206A V208A N211A
mel6	K263A K265A E2661 N267A
mel7	Y301A N302A K304A E307A
Mel8	E322A R324A V326E N329A

## Data Availability

The original contributions presented in this study are included in the article/Appendix A. Further inquiries can be directed to the corresponding author(s).

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
