# Peer review of "EPY001, a Novel Monoclonal Antibody Against Pseudomonas aeruginosa Targeting OprF"

_ijms, 2025, doi:10.3390/ijms262110380_

Round 1
Reviewer 1 Report
Comments and Suggestions for Authors
The manuscript title clearly indicates a focus on EPY001 and its target, OprF.
The abstract is well-structured but could be improved by briefly mentioning the significance of OprF in the pathogenesis of Pseudomonas aeruginosa, which would provide better context for readers.
In the introduction, a clearer transition between the clinical significance of P. aeruginosa and the rationale for using monoclonal antibodies as a therapeutic strategy would enhance the flow of ideas.
It is important to ensure consistent use of abbreviations throughout the manuscript; for instance, "P. aeruginosa" should be clearly introduced upon its first mention and used consistently thereafter.
A thorough proofreading is recommended to correct minor typographical and grammatical errors, which will enhance overall readability.
After making these minor revisions, the manuscript should be suitable for acceptance and publication.
Author Response
Dear Reviewer,
We sincerely thank you for taking the time to review our work. Your comments are highly valuable and will help us significantly improve the quality of this manuscript.
If the article is accepted, the review process will be available to all readers, enhancing the transparency of our work. Although we were not able to demonstrate the in vivo efficacy of the EPY001 antibody, we believe that this three-year study is important to share with the scientific community, as it may help explore new directions in the fight against antibiotic resistance.
We are very grateful for your constructive comments and the attention you have given to our manuscript.
Comments and Suggestions for Authors
The manuscript title clearly indicates a focus on EPY001 and its target, OprF.
Comments 1 : The abstract is well-structured but could be improved by briefly mentioning the significance of OprF in the pathogenesis of Pseudomonas aeruginosa, which would provide better context for readers.
Response 1 : Thank you for pointing this out. Therefore, we have updated the abstract. See Line 18.
« OprF, the major and highly conserved outer membrane protein of Pseudomonas aeruginosa, plays key roles in the pathogenesis of this bacterium, including biofilm formation, host cell adhesion, immune sensing, and resistance to macrophage clearance, making it a crucial factor in virulence and a promising immunotherapeutic target. »
Comments 2 : In the introduction, a clearer transition between the clinical significance of P. aeruginosa and the rationale for using monoclonal antibodies as a therapeutic strategy would enhance the flow of ideas.
Response 2 : Thank you for this useful comment. Therefore, we included a transition sentence. See Line 48.
« Given the rising threat of antibiotic-resistant P. aeruginosa and the limitations of existing treatments, there is a pressing need for alternative targeted strategies that can specifically neutralize the pathogen while minimizing side effects ».
A more detailed explanation can be found in the introduction regarding why monoclonal antibodies (mAbs) are promising:
- The growing problem of antibiotic resistance to currently available drugs.
- The lack of new antibiotics against Gram-negative bacteria with novel mechanisms of action for several decades.
- The specificity of monoclonal antibodies reduces side effects and spares the commensal flora.
- Some monoclonal antibodies are already used to treat bacterial infections.
- Many other monoclonal antibodies are currently under development.
Comments 3 : It is important to ensure consistent use of abbreviations throughout the manuscript; for instance, "P. aeruginosa" should be clearly introduced upon its first mention and used consistently thereafter.
Response 3 : We have updated the manuscript.
Reviewer 2 Report
Comments and Suggestions for Authors
Review report/comments
The authors develop a novel P. aeruginosa antibody which targets outer membrane protein - OprF, one of the most abundant proteins on the Pseudomonas surface. The authors produced and analyzed a novel antibody targeting OprF which confers very high affinity (picomolar range) towards the target protein both in proteoliposomes and live bacterial tests. EPY001 successfully reduced the formation of biofilms in one P. aeruginosa strain, as well as a slight reduction pyocyanin production in the same strain. Unfortunately, the were no successful signs of in vivo inhibition or P. aeruginosa infection (in mice models nor macrophage systems) which was the main aim of the paper. The authors additionally gained a more detailed insight of which protein segment the antibody recognizes (loop 5 with contributions of two other outer membrane loops), which is useful for future developments of high potency antibodies. Overall, this work provides useful insight into the OprF significance in some bacterial processes, such as differential antibiotic susceptibility, and adds very early steps in development of surface targeting antibodies on live P. aeruginosa bacteria which could be further developed in the future.
Major remarks:
- I suggest plotting the Kd values obtained in the initial binding assays using the average value and std from triplicate experiments, rather than simply reporting it in the text as an approximate range (lines 131, 132).
- Since the main aim of this work: production of the inhibitory antibody to P. aeruginosa has not been fully achieved (in vivo testing, macrophage activation), the authors should expand on the possible reasons why such high affinity antibody fails to work in in vivo setting in the Discussion.
- While mice experiments can fail to work due to many factors of in vivo systems, the authors should provide more explanations in the discussion as to why phagocytosis would not be promoted using macrophages in ADCP experiments. Based on the affinity values, and titer used, the bacteria would be almost fully coated with EPY001, yet there is no significant increase in phagocytosis by macrophages. Does this mean that aside from a high affinity epitope interaction, the rest of the EPY001 antibody is not functional to activate phagocytosis?
- in general I am surprised that all statistical tests (Figures 2, 5, 6) yield 0.03 for p-value. Can this be presented as the exact number from the MW statistical test (up to 3rd decimal)?
Minor remarks:
- Line 398: “Gentamycin acetyltransferases specifically modify aminoglycosides by adding an acetyl group, which prevents the antibiotic from binding to its target, the bacterial ribosome.” Add citation that supports this.
- Line 181 – the Figure reference should be Figure 3a, and not 2a if referring to the following figure.
- Fig. 3a and 3b – there is a red line below the “survival” word on the y-axis. This should be removed.
- E. coli was first introduced in line 98 in abbreviated form and then in line 144 as a full name. This should be reversed.
- the bacterial strain P. aeruginosa is written mixed in italic and regular font (e.g., lines 128, 137 vs 149). This should be uniform (always italic) throughout the entire manuscript. This includes other bacteria such as A. baumannii and E. coli.
- In vivo and in vitro are written mixed in italic and regular font throughout the paper. This should be uniform and in italic (see lines 210, 211 for example).
Reviewer 3 Report
Comments and Suggestions for Authors
Report
Title: EPY001, a novel monoclonal antibody against Pseudomonas aeruginosa targeting OprF
The goal of this work is to evaluate the activity of EPY001, a monoclonal antibody specific to OprF, a conserved outer membrane protein found in P. aeruginosa and to qualitatively assess the functional characterization of EPY001 activity on biofilm formation, pyocyanin production, and antibiotic resistance. It is a very interesting study and it is clear the authors have put in much effort especially in the experimental part. Also the work is comprehensive and well-designed. Nevertheless, I think there are number of points (listed below) that could be clarified or revised in order to improve the overall clarity and impact of the manuscript.
Major concerns
- Results
Line 226, authors stated that the absence of OprF in strain CHA resulted in increased resistance to ticarcillin and piperacillin. but also i see the same increase in resistance in Figure 4a, in Ticarcillin/clavulanic acid.
Line 263, authors mensioned that the experimental replicates was four. please add this statement in the related part in the materials and methods.
Line 292, in Evaluation of Complement-dependent cytotoxicity assay Activity of EPY001 on the 292 CHA Strain. Why authors only test the CHA strain in this experiment?
Line 294, authors should write the full term out at its first occurrence with the abbreviation in parentheses. After the first instance, you must use the abbreviation throughout the manuscript.
- Discussion
- Line 341, its better to start the discussion paragraph with a brief summary of what done in this study, and also why
- The discussion section is very good, and it covers almost all the points, but it is considerably long when compared to a typical manuscript of this type. I suggest shortening, and tightening the discussion. I believe that if you condense and tighten your discussion it will be easier for the reader to follow and keep their attention.
- Materials and Methods
Line 596, In this experiment, the authors utilized a concentration of liposomes in a range of 1–4 mg/mL. Could you please explain why you selected this range?
Line 617, the lead candidate is EPY001. But in lines 628, 634, and 643 EPY0001 was mentioned. Is this the same or different one? Please revise and clarify
Line 673, in Animal investigation protocol, what is the acclimatization period of mice?
Line 701, in Antibiotic Susceptibility Testing, authors didn’t mention the number of replicated used.
Line 701, what are the positive and negative control used in the Antibiotic Susceptibility Testing?
Line 708, authors used several commercial antibiotic disks (Ticarcillin, Piperacillin, Levofloxacin and Piperacillin/Tazobactam, Oxoid, Thermo 708 Fisher). Please provide the concentration of each antibiotic disk used.
Line 711, please provide the company names of powder antibiotics (which used in MIC), as well as the antibiotic concentration range used in the experiments.
Line 716, in Biofilm formation assays. authors should state the ability of the used Pseudomonas strains to form a biofilm, and also categorize it (strong, moderate or weak).
Line 716, in Biofilm formation assays, authors didn’t mention the number of replicated used.
Line 725, what are the positive and negative control used in the biofilm formation assay?
Line 728, in Pyocyanin production assays. authors didnt include the use EPY001 and its concentration as anti-pyocyanin.
Line 729, in Pyocyanin production assays. what is the bacterial inoculum/concentration used to inoculate the 5ml LB?
Line 731, author state that “The amount of pyocyanin was estimated by the ratio of the absorbance measured at 520 and the absorbance measured at 600 nm.” please clarify the measurement of the absorbance at 600 nm beside 595nm?
Line 743, in Complement Dependent cytotoxicity (CDC) Activation Assay. Is the 40% is fixed for both complements, or what it used for?
Line 744, how many replicates done for CFU counting?
Line 770, Antibody-Dependent Cellular Phagocytosis (ADCP) Activation Assay. what is the total volume of the final reaction components?
- Conclusions
Line 775, Conclusions. there are two abbreviations in the conclusion section which are CDC and ADCP. In conclusion authors should remove all abbreviations and only full terms should be provided.
Minor concerns
Abstract
Line 14. Pseudomonas aeruginosa must be written in italic
- Introduction
Line 33. Writing mistake “Salmonella tiphy”, should be written “Salmonella typhi”
Line 34. Writing mistake “Enterococcos”, should be written “Enterococcus”
- Results
Line 128, Line 137, Line 143, and Line 144, all bacteria names must be written in italic “P. aeruginosa, Acinetobacter baumannii and a Escherichia coli”
Line 144, Remove the letter a before Escherichia coli
Line 149, please remove this sentence from the results section "OprF is an outer membrane protein of P. aeruginosa that presents eight extracellular loops, each potentially accessible to antibody binding [36]. "
Line 153, please provide the number of the related materials and methods sub-heading.
Line 166, writing mistake "loos"
Line 184, i suggest the vertical arrangement (one top of the other) of Figure 3 panels a, b and c, it will be clearer for viewer's vision.
Line 208, Line 210, and Line 211, in vivo and in vitro must be written in italic.
Line 214, Line 239, Line 241. P. aeruginosa should be written in italic.
Line 220, also the same as Figure 3, i suggest the vertical arrangement in Figure 4.
- Materials and methods
Line 706, written mistake "0.5 McFarland" not “O.5 McFarland”
- Note: there is no Appendix file.

Comments on the Quality of English Language
Require some improvement
Round 2
Reviewer 2 Report
Comments and Suggestions for Authors
My comments have been sufficiently addressed.
Author Response
We thank the reviewer for the insightful comments and suggestions that strengthened our manuscript.
Reviewer 3 Report
Comments and Suggestions for Authors
Report (Decision: Accept)
Thank you for the opportunity to evaluate this manuscript “EPY001, a novel monoclonal antibody against Pseudomonas aeruginosa targeting OprF” for your journal, and I would like to also acknowledge the authors’ significant work and effort on the manuscript. The authors have really improved the manuscript based on the comments provided earlier, and it is now clearer and more consistent, and acceptable for publication.
I have 2 minor points:
- The authors state that statistical analyses were conducted (as indicated in the figure captions), but the relevant methods are not mentioned in the text. Please add a separate subsection to describe the statistical analysis at the end of the Materials and Methods section.
- Please be sure that all bacterial names in the References section are in italics, as is customary in scientific nomenclature.
